

# Vertical and horizontal distribution of sub-micron aerosol chemical composition and physical characteristics across Northern India, during the pre-monsoon and monsoon seasons

James Brooks[1], James D. Allan[1,2], Paul I. Williams[1,2], Dantong Liu[3], Cathryn Fox[4], Jim Haywood[4,5],

Justin M. Langridge[4], Ellie J. Highwood[6], Sobhan K. Kompalli[7], Debbie O'Sullivan[4], Suresh S. Babu[7],

Sreedharan K. Satheesh[8], Andrew G. Turner[3,6], Hugh Coe[1].

[1] Centre for Atmospheric Science, School of Earth and Environmental Sciences, University of Manchester, Manchester, UK.

[2] National Centre for Atmospheric Science, UK.

[3] School of Earth Sciences, Zhejiang University, China.

[4] Observation Based Research, Met Office, Exeter, UK.

[5] College of Engineering, Mathematics & Physical Sciences, Exeter, UK.

[6] Department of Meteorology, University of Reading, UK.

[7] Space Physics Laboratory, Vikram Sarabhai Space Centre, India.

[8] Centre for Atmospheric & Oceanic Sciences, Indian Institute of Science, India.

*Correspondence to*: Prof. Hugh Coe (hugh.coe@manchester.ac.uk)



**Abstract.**

The vertical distribution in the physical and chemical properties of submicron aerosol has been characterised across northern India for the first time using airborne in-situ measurements. This study focusses primarily on the Indo-Gangetic Plain, a low-lying area in the north of India which commonly experiences high aerosol mass concentrations prior to the monsoon season. Data presented are from the UK Facility for Airborne Atmospheric Measurements BAe-146 research aircraft that performed flights in the region during the 2016 pre-monsoon (11$^{th}$ and 12$^{th}$ June) and monsoon (30$^{th}$ June to 11$^{th}$ July) seasons.

Inside the Indo-Gangetic Plain boundary layer, organic matter dominated the submicron aerosol mass (43%) followed by sulphate (29%), ammonium (14%), nitrate (7%) and black carbon (7%). However, outside the Indo-Gangetic Plain, sulphate was the dominant species contributing 44% to the total submicron aerosol mass in the boundary layer, followed by organic matter (30%), ammonium (14%), nitrate (6%) and black carbon (6%). Chlorine mass concentrations were negligible throughout the campaign. Black carbon mass concentrations were higher inside the Indo-Gangetic Plain (2 µg/m$^3$ std) compared to outside (1 µg/m$^3$ std). Nitrate appeared to be controlled by thermodynamic processes, with increased mass concentration in conditions of lower temperature and higher relative humidity. Increased mass and number concentrations were observed inside the Indo-Gangetic Plain and the aerosol was more absorbing in this region, whereas outside the Indo-Gangetic Plain the aerosol was larger in size and more scattering in nature, suggesting greater dust presence especially in northwest India. The aerosol composition remained largely similar as the monsoon season progressed, but the total aerosol mass concentrations decreased by ~50% as the rainfall arrived; the pre-monsoon average total mass concentration was 30 µg/m$^3$ std compared to a monsoon average total mass concentration of 10-20 µg/m$^3$ std. However, this mass concentration decrease was less noteworthy (~20-30%) over the Indo-Gangetic Plain, likely due to the strength of emission sources in this region. Decreases occurred in coarse mode aerosol, with the fine mode fraction increasing with monsoon arrival. In the aerosol vertical profile, inside the Indo-Gangetic Plain during the pre-monsoon, organic aerosol and absorbing aerosol species dominated in the lower atmosphere (<1.5 km) with sulphate, dust and other scattering aerosol species enhanced in an elevated aerosol layer above 1.5 km with maximum aerosol height ~6 km. As the monsoon progressed into this region, the elevated aerosol layer diminished, the aerosol maximum height reduced to ~2 km and the total mass concentrations decreased by ~50%. The dust and sulphate-dominated aerosol layer aloft was removed upon monsoon arrival, highlighted by an increase in fine mode fraction throughout the profile.





# 1 Introduction

South Asia is one of the world's most populous and fastest growing regions, with 24% of the world's population. The diverse living habits, fuel and land use make understanding the complexity of emissions and atmospheric dynamics challenging. The Indo-Gangetic Plain (IGP) in the Indian sub-continent is one such polluted region in South Asia; its unique geomorphology, meteorology and characteristic variations in aerosol particles have drawn the attention of aerosol researchers for many years (Pawar et al., 2015; Singh et al., 2017). Aerosol mass sources within the IGP are mostly dominated by natural sources such as dust, but also anthropogenic emissions associated with traffic, biomass burning, waste incineration and crop and wood burning (Banerjee et al., 2015; Singh et al., 2017), with Delhi being one of the most polluted locations (Sharma et al., 2014).

Aerosol modelling studies have shown that aerosol present over India can significantly influence the monsoon through its role in redistribution of heat (Satheesh and Ramanathan, 2000; Ramanathan et al., 2001; 2005; Nair et al., 2012; 2013). Modelling comparisons have shown considerable uncertainty in model predictions of regional heating from aerosols, and hence their impact and characteristics with regards to the Indian monsoon (Myhre et al., 2013). In order to improve current scientific knowledge and challenge global chemical transport and climate models, in-situ measurements of sufficient accuracy and sensitivity are required to assess the key parameters such as mass, optical and chemical properties of the ambient aerosol. In India, systematic investigations into the physio-chemical properties of aerosol, their temporal heterogeneities, spectral characteristics and size distributions throughout the atmospheric column have been subject to analysis for several decades (Satheesh and Ramanathan, 2000; Ramanathan et al., 2001; 2005; Moorthy et al., 2004; 2009; 2013; 2016; Singh et al., 2006; Babu et al., 2011; Gautam et al., 2011; Samset et al., 2012). However, lack of in-situ analysis has hampered progress in reducing uncertainty behind the current scientific understanding.

Much uncertainty surrounds the vertical structure, spatial and seasonal extent, and heterogeneous chemistry of aerosol over India (Pan et al., 2015). There are only a few studies that address the physical properties of the aerosol by analysing the single scattering albedo (SSA) (Ganguly et al., 2005; Sreekanth et al., 2011; Babu et al., 2016; Vaishya et al., 2018). One reason is in part due to uncertainties in characterising aerosol absorption accurately. For northern India in spring it appears that the SSA within the boundary layer is relatively low, although increasing with altitude in the atmospheric column (Babu et al., 2016). This is possibly indicative of dust transport at high altitudes from regions such as the Rajasthan Desert in the northwest of the country. Sarangi et al. (2016) was to an extent able to elaborate on these findings and found that the elevated aerosol layer was a mix of coarse-sized natural and fine-sized anthropogenic aerosol, both influenced by local emission and long-range transport, yet they highlight the requirement for more in-situ chemical composition analysis in the atmospheric column.

A handful of efforts have been made to explore the sources, variability and chemical composition of the ambient aerosol across northern India by using an aerosol mass spectrometer (AMS) at a number of surface sites (Bhattu and Tripathi, 2015; Chakraborty et al., 2015; 2016; Kumar and Yadav, 2016; Kumar et al., 2016). These studies have covered both winter and summer seasons in northern India, finding that aerosols are dominated by organic matter followed by inorganic



components and black carbon. These previous AMS studies have provided a detailed insight into aerosol chemical composition for the built-up region of Kanpur in the Indo-Gangetic Plain. However, regional variations in the aerosol chemical composition using such detailed methods have not been carried out across this region to date.

This paper presents the first airborne measurements of aerosol chemical characteristics over northern India. The vertical distribution of aerosol chemical composition and physical properties is a major focus of this study, along with understanding how the monsoon progression across the region influences the aerosol characteristics.

## 2 Methodology and Climatology

Twenty-two science flights were conducted by the UK Facility for Airborne Atmospheric Measurement (FAAM) BAe-146 research aircraft with the flight tracks highlighted in Figure 1 and flight summaries in Table 1. The flights took place during two periods: the pre-monsoon (11th and 12th June 2016) and the monsoon onset period (30th June to 11th July 2016), based at Lucknow (LKN; 26.85°N, 80.95°E). The aircraft flew with a comprehensive instrument suite, capable of measuring aerosols, cloud physics, chemical tracers, radiative fluxes and meteorological fields, however only instruments used in this analysis are discussed further. The FAAM BAe-146 has a typical range of ~3000 km and an altitude ceiling of over 10 km, with an aircraft science speed of ~100 ms$^{-1}$. From the operating base, the aircraft typically covered radial distances of ~200-300 km in 4.5/5 hours of flight time, resulting in over 120 hours flying completed throughout the campaign (89 hours science). Based on the likely synoptic and local conditions on the day, different types of science flights were conducted; radiation flights and survey flights. Both flight types consisted of long-leg duration flights covering the NE Bay of Bengal and Indo-Gangetic Plain (IGP) regions, delivering the main part of the aerosol characterisation. In addition, profiles to high altitudes when taking off from Lucknow and in selected other locations were carried out in order to build up a statistical picture of the vertical structure. Low altitude straight-level runs (SLRs) were also carried out at heights of around 0.5-1.0 km.

### 2.1 Instrumentation

The compact Time-of-Flight Aerosol Mass Spectrometer (cToF-AMS) provided the capability to quantitatively measure the size-resolved chemical composition of non-refractory submicron particle matter (NR-PM1) organic aerosol (OA), sulphate (SO$_4$), ammonium (NH$_4$), nitrate (NO$_3$) and chloride (Cl) (Morgan, 2010). A major advantage of using the cToF-AMS is the ability to provide high temporal resolution measurements with enhanced precision and sensitivity, therefore making it suitable for aircraft operation. In the rest of the paper we refer to the cToF-AMS as "AMS" for brevity. The AMS data were acquired in mass spectrometer (MS) mode, which produces species concentrations and a complete mass spectrum of the non-refractory submicron mass with no size information, but with higher sensitivity than the particle time-of-flight (PToF) mode. Previous studies (Crosier et al., 2007; Morgan et al., 2010) have detailed the AMS sampling strategy onboard the aircraft. The sampling losses for the AMS inlet system were estimated to be approximately 10% by number across the size-range measuring capability of the AMS. The system ammonium nitrate residence time is sufficiently short compared to timescales for mass





transfer that repartitioning of ammonium nitrate towards the gas phase is small. The AMS was connected to a Rosemount inlet (Foltescu et al., 1995) via stainless steel tubing with a total residence time of several seconds for the entire inlet system. Samples were introduced at ambient pressure and sub-micron particle losses are considered negligible for the operating altitudes considered by this study (Trembath et al., 2012). Furthermore, wing tip to wing tip comparisons between different

AMS instruments using different inlet setups have shown less than 15% difference (Crosier et al., 2007). The aerosol introduced to the AMS is considered dry, due to the coupled effect of the aircraft cabin temperature (~300 K) and additional heating due to the decelerating sample air flow (Morgan et al., 2010). Power was unavailable between flights due to operational constraints but through the use of plug valves to isolate the AMS chamber, a vacuum of typically less than 0.5 Torr was maintained whilst the turbo-pumps were powered down.

A Single Particle Soot Photometer (SP2) manufactured by Droplet Measurement Technologies (DMT) Inc. (Boulder, CO, USA) was used to provide measurements of refractory black carbon (BC) (Stephens et al., 2003; Baumgardner et al., 2004; Liu et al., 2010). The instrument operation and data interpretation procedures are described elsewhere (Liu et al., 2010; McMeeking et al., 2010). The SP2 incandescence signal was calibrated for rBC mass using Aquadag® black carbon particle standards (Aqueous Deflocculated Acheson Graphite, manufactured by Acheson Inc., USA) and corrected for ambient rBC

with a factor of 0.75 (Laborde et al., 2012). Reported mass loadings have a measurement uncertainty of approximately 30% (Schwarz et al., 2008; Shiraiwa et al., 2008; Liu et al., 2014).

Aerosol number concentrations were measured using a Passive Cavity Aerosol Spectrometer Probe (PCASP) and a Cloud Droplet Probe (CDP). The PCASP measured both aerosol number and size in the range of 0.3-3 µm. It is an optical particle counter (OPC) and was fitted to the underside of the aircraft wings. The CDP collected droplet size measurements

between 3-50 µm (cloud cleared). Data provided by these two instruments allow calculation of the aerosol fine mode fraction, which indicates whether large (super-micron) or small (sub-micron) aerosol dominates the number concentrations (Crosier et al., 2011; Cai et al., 2013). The aerosol size distributions have been calibrated to the refractive index of the ambient aerosol, based on the fractional composition during each flight.

Aerosol single scattering albedo (SSA) values at 550nm at standard temperature and pressure (STP) were calculated

using Nephelometer (TSI Model 3563) and Particle Soot Absorption Photometer (PSAP) (designed by Radiance Research) data. Scattering measurements were provided from the nephelometer, with continuous particulate absorption measurements made by the PSAP. The nephelometer was corrected using the Muller correction scheme (Muller et al., 2011) and the PSAP via the Bond et al. (1999) corrections. More information on these instruments and calculations can be found in Haywood and Osborne (2000).

Carbon monoxide mixing ratios were measured by an Aero-Laser A15002 VUV resonance fluorescence gas analyser (Gerbig et al., 1999), where total uncertainty of the FAAM instrument is estimated to be 2% (O'Shea et al., 2013). In flight calibrations were performed using World Meteorological Organisation traceable gas standards.



A General Eastern 1011B (GE Measurement & Control) chilled hygrometer provided measurements of ambient dew point temperature, accurate to 0.2 °C. A Rosemount/Goodrich type-102 true air temperature sensor was mounted outside the aircraft, providing ambient temperature measurements using a Rosemount 102AL platinum resistance immersion thermometer.

A Leosphere ALS450 elastic backscatter lidar (wavelength 355 nm) was deployed on the FAAM aircraft in a nadir-viewing geometry. Marenco et al. (2011) and Marenco (2013) describe the methodology for converting lidar beam returns at 355 nm wavelength into profiles of aerosol extinction coefficient. The system specifications are summarised in Marenco et al. (2014) and references therein, and a further description of the data processing methodology can be found in Marenco et al. (2016). During processing, the lidar data was integrated to 1-minute temporal resolution, which corresponds to a $9 \pm 2$ km footprint at aircraft speeds. Smoothing to a 45 m vertical resolution was also applied to reduce the effect of shot noise. The vertical profiles were processed using a double iteration. First, we determined the lidar ratio (extinction-to-backscatter ratio) and subsequently processed the full data set to determine the extinction coefficient (see Marenco et al. (2016) and references therein for details). As in the above-mentioned papers, the first iteration was conducted on a subset of the vertical profiles, where the signature of Rayleigh scattering above the aerosol layer could clearly be identified to enable the lidar ratio to be determined. A mean campaign lidar ratio of $46 \pm 10$ sr was obtained, which is in good agreement with other measurements of the lidar ratio for smoke and mixed aerosol at 355 nm (Muller et al.., 2007). This value of the lidar ratio was then used to process the whole dataset in the second iteration. For dust during this campaign, it is estimated that the uncertainty in the derived extinction was 11%. The uncertainty is smaller than this near the top of the profile (closer to the aircraft) and larger nearer the ground.

## 2.2 AMS data quantification

The AMS data analysis was performed using the standard SQUIRREL (SeQUential Igor data RetRiEvaL) ToF-AMS software package, with mass spectrum deconvolution accomplished using the fragmentation table approach described by Allan et al. (2004). Error estimates were generated according to the model documented by Allan et al. (2003) and mass concentrations derived from the AMS are reported as micrograms per standard cubic metre ($\mu g/m^3$) i.e. at a temperature of 273.15 K and pressure of 1013.25 hPa. Ionisation efficiency (IE) calibrations were performed regularly before and after each flight during the flying period, with values determined from both pre-flight and post-flight calibrations (i.e. taking place on the same day) usually exhibiting little variability. Post-flight values were used as these were considered to be more reliable compared to pre-flight due to reduced instrument background post-flight. During B969 (see flights included in this analysis in Table 1) a failure with the filament supplying ions to the reagent gas means the data from that flight is erroneous and is not included in this analysis. Procedures for AMS calibration can be found in previous publications (Allan et al., 2003; 2004; Jiminez et al., 2003). The amplification factor of the microchannel plate detector (MCP) was measured during every pre-flight and most post-flight calibrations, when the AMS was in use ("single ion calibration"). Due to varying flight conditions, single ion values were taken for each flight rather than using one value for the entire campaign. The particle collection efficiency (CE) for the AMS was determined using the approach of Middlebrook et al. (2012).



## 2.3 Indian Summer monsoon

The Indian monsoon is one of the world's most dramatic seasonal variations in climate, characterised by the reversal of prevailing winds between winter and summer. The summer monsoon typically lasts from June to September and supplies India with 80% of its rainfall (Lau and Yang, 1997) and at each location, the monsoon arrival brings cooler and moister conditions to replace the pre-monsoon heat (Parker et al., 2016). The monsoon onset typically occurs around 1st June, in Kerala on the southwest coast, and then progresses north arriving in Delhi before the end of July. Dates of the 2016 summer monsoon progression can be found in appendix Figure A1, with the 2016 summer monsoon arriving in Lucknow, for example, on the 1st July at around 2 weeks later than the climatological average.

Mean synoptic meteorological conditions for our flying period (see Figure 2) at the surface level and aloft over the IGP in the vicinity of Lucknow can play a significant role in the modulating aerosol properties over northern India (Li et al., 2016; Parker et al., 2016; Sen et al., 2017). Figure 2 highlights the meteorological averages from the days that flying occurred, with pre-monsoon (11th - 12th June) and monsoon (30th June - 11th July) flights. The IGP typically experiences sub-tropical climatic conditions, with very high near-surface temperatures of ~305-315 K during the pre-monsoon. In the pre-monsoon, the wind pattern in the boundary layer is south-westerly/westerly bringing warm, dry air from arid regions such as the Rajasthan Desert, Pakistan and beyond. This dry, hot air brings with it high aerosol loadings, and coupled with the strong emissions sources within the IGP, causes high aerosol concentrations to build up in the Himalayan foothills and across the IGP and northern India region. This is similar above the boundary layer also, at pressure altitudes ~650 hPa, with dry arid aerosol laden air being transported from NW India and beyond into the IGP. In the pre-monsoon the near-surface air temperatures are at their peak of ~310 K, near-surface RH is low ~60%, and soil moisture and precipitation are low. As the monsoon progresses from the south, however, the wind patterns change; in the boundary layer, the wind direction shifts to a south-easterly, advecting cooler, moister air masses to replace the dry, hot dry desert air that dominated during the pre-monsoon. The increasing dominance of south-easterly winds in the boundary layer increases the supply of moisture from the Bay of Bengal for the central IGP region, with moisture influx from the Arabian Sea to the NW region over Jaipur and Jodhpur.

During the 2016 monsoon, in early July, synoptic conditions were perturbed by the passage of a monsoon depression (a strong low-pressure system) passing from the Bay of Bengal across the IGP. Its cyclonic signature can clearly be seen in Figure 2C and D. The wind direction in the mid-troposphere also changes as the monsoon develops, with a gradual shift in wind direction from westerlies to south-easterlies. The dry air aloft is gradually moistened from below, becoming progressively cooler and more humid. Parker et al. (2016) explain that at this time the equivalent potential temperature increases, favouring the formation of clouds with monsoon progression. As the first monsoon rainfall arrives, the soil is dry, so boundary layer moisture is advected from the ocean. For subsequent rainfall, moisture can also be advected from the land surface, moistening the boundary layer and leading to the development of deep cumulus cloud. The boundary layer RH increases to upwards of 100% (and supersaturated air can be found (>100% RH)) and the temperatures decrease to ~295-300 K. Just as in the case of




the boundary layer air, flow during the mature monsoon into July 2016 is modulated at 650 hPa (Figure 2D) by the presence of a monsoon depression.

## 3 Results

The pre-monsoon (11th – 12th June, flights of B956 and B957) and monsoon (30th June – 11th July, flights of B968-B976) are considered separately. Of this second category, the first few flights can be regarded as occurring during the monsoon transition phase, since the monsoon has arrived at some locations but not others. For example, during B968 (30th June) the monsoon was seen to be influencing the IGP but not Jaipur in NW India. More information on the monsoon development can be seen in Figure 2 and appendix Figure A1 (in this example for 30th June, the monsoon progression isochrone for 2016 still lies to the south and east of Jaipur). The analysis presented below first outlines the pre-monsoon measurements of the aerosol chemical composition and then physical characteristics, followed by the changes seen as the monsoon progressed, for the boundary layer (Sect. 3.1) and vertical profile (Sect. 3.2).

The aerosol burden over northern India has a distinct structure in the pre-monsoon, both in terms of aerosol species and thermodynamic nature, with large changes as the monsoon progresses. Figure 3 highlights the pre-monsoon structure of aerosol extinction coefficient obtained from lidar data. The data show an elevated aerosol layer between 3-6km across northern India, particularly over NW India with somewhat decreased amounts over NE India, during the pre-monsoon. The boundary layer aerosol over the NW shows increased levels of aerosol extinction, with these high extinction values present in the IGP and over NE India. The thermodynamic profiles in Figure 3 for the various locations show differences between the boundary layer and aloft across northern India; heights that coincide with elevated aerosol have consistent high RH values across locations, with large variations in RH in the boundary layer. It is clear from the lidar data that there is distinct structure in aerosol vertical profiles.

Considering the above, an aerosol composition summary in Figure 4 highlights the pre-monsoon and monsoon structure of the aerosol vertical profile over northern India. The key measurements needed over this region of relevance to the testing of general circulation models (GCMs) include aerosol size information as a function of height, composition, scattering and absorption characteristics. In order for models to correctly test for accurate aerosol representation, these criteria need to be met. The aerosol vertical summary plot highlights large aerosol loadings upwards of ~50 µg/m$^3$ present in the pre-monsoon for all locations across northern India, with the distinct elevated aerosol layer present over the IGP and NE India dominated by larger particles between 3-7 km. As the monsoon progresses the aerosol concentrations decrease in both total mass and maximum height, alongside decreases in aerosol scattering coefficient and aerosol size. Black carbon is only seen in significant mass concentrations inside the BL in both the pre-monsoon and monsoon seasons.



## 3.1 Boundary layer aerosol during the progression of the monsoon

Summary statistics of boundary layer aerosol chemical composition for the various regions in northern India are presented in Figure 1, and more detailed analysis of the aerosol chemical composition for Straight Level Runs (SLRs) below 1000 m shown in Figures 5 and 6. The boundary layer height was predominantly above 1500 m, such that the SLR data

presented here are within the boundary layer for all locations.

During the pre-monsoon flights (B956+B957), aerosol average total mass concentrations were at their highest inside the IGP compared to outside, and there were also differences in the particle composition. In the pre-monsoon, boundary layer total mass concentrations were ~50 µg/m$^3$ (±10 µg/m$^3$) inside the IGP with total boundary layer mass concentrations ~15 µg/m$^3$ (±5 µg/m$^3$) outside the IGP. Figures 5 and 6 highlight the regional variations in the dominant aerosol species present over the

various locations in northern India. Organic aerosol dominated inside the IGP, with average mass concentrations present ~12 µg/m$^3$ comprising 43% of the typical mass fraction, with SO$_4$ at ~10 µg/m$^3$ comprising 29%. However, outside the IGP, the dominant species in the boundary layer was SO$_4$ with average mass concentrations ~10 µg/m$^3$, comprising 44% of total aerosol loading, whereas the organics ~6 µg/m$^3$ comprising 30%. Organic aerosol underwent large changes in mass concentration dependent upon location with mass concentrations inside the IGP twice those measured outside, whereas SO$_4$ displayed

relatively consistent values across northern India. NH$_4$ mass concentrations displayed similar patterns to the SO$_4$ regional burden, indicating the predominance of ammonium sulphate aerosol across northern India. Nitrate, on the other hand, was not present in significant concentrations during the pre-monsoon regardless of location (0-1 µg/m$^3$).

As the monsoon progressed over northern India, it developed from the south east towards the north west, as shown in Figure 2 and appendix Figure A1. Inside the IGP, the average total mass concentration decreased from 25-35 µg/m$^3$ to 15-25

20  µg/m$^3$. Outside the IGP however, the average total mass concentration was similar to the pre-monsoon value of ~15 µg/m$^3$. During the eastern flights between Lucknow and Bhubaneswar, as the monsoon arrived and developed during mid-June (B971 and B975) a decrease in aerosol mass concentration occurred from 10-30 µg/m$^3$ to 1-10 µg/m$^3$. Organic aerosol decreased in mass concentration as the monsoon progressed, especially over NW and NE India but not inside the IGP. Similar changes in sulphate and ammonium aerosol mass concentrations were also observed. Compared to pre-monsoon concentrations, nitrate

aerosol mass concentration increased in contrast to the decrease in the concentration of other species. When rainfall arrived in northern India, nitrate concentrations increased from 0-1 µg/m$^3$ to 5-10 µg/m$^3$ in the IGP. Increased concentrations were also observed outside the IGP. This finding indicates not only that nitrate aerosol is strongly driven by thermodynamic variables (increased relative humidity and soil moisture, and decreased air temperatures), but also that as the humid and cooler air arrived there was sufficient ammonium to neutralise the nitric acid present and partition a significant fraction of this to the particle

phase within the boundary layer under these conditions.

The aerosol physical properties, shown in Figures 7 and 8, also vary between the IGP and other regions in northern India. During the pre-monsoon, aerosol number concentrations for both coarse and accumulation mode particles were high. Accumulation mode number concentration was greatest within the IGP boundary layer, coinciding with high organic aerosol





loading in the IGP region and consistent with the significantly elevated rBC observations. To the west of the IGP however, the boundary layer had a higher proportion of coarse mode aerosol. This suggests the dominance of dust and other scattering aerosol species in these regions, consistent with the land surface type (dry, arid deserts) and the dominance of sulphate aerosol outside the IGP. These findings are consistent with the size distributions presented in Figure 8. During the pre-monsoon, larger

particles dominate the aerosol volume size distribution both inside and outside the IGP, as both regions are influenced by the prevailing wind direction from the arid, dusty regions in the NW of India. As the monsoon progresses, Figure 8C indicates the removal of larger particles particularly inside the IGP, consistent with the change in prevailing wind pattern in Figure 2.

As with the aerosol chemical composition, the aerosol physical properties underwent changes during the monsoon progression. After the monsoon rains had arrived, the fine mode fraction within the IGP increased, indicating that larger aerosol

particles (such as suspended dust) are being washed out by the monsoon and that increased surface wetness supresses the suspension of dust (as shown in Figures 7 and 8). Outside the IGP towards NW India, the aerosol physical properties remained similar to the pre-monsoon conditions, with larger aerosol particles dominating. This is partly due to the arrival of the monsoon in this region, the last region to receive the rains. Thus, during B976 (final science flight of the campaign) the monsoon had not fully developed in the far NW, resulting in little monsoon rain washout. The size distribution information in Figure 8 agrees

with this, with a decrease in amount of larger aerosol inside the IGP with monsoon arrival there, whereas outside the IGP the larger particle concentrations remain similar to the pre-monsoon. Spatial changes in the aerosol scattering properties were also witnessed with monsoon progression. When the surface wetness is increased with monsoon arrival, dust suspension can be prevented but anthropogenic emissions can continue, so once the rain has removed the particulates in the atmosphere and sources of aerosol change, then the atmospheric burden will be altered. Also, with monsoon arrival the origins of air masses

change, which may have played a role in the decreasing presence of large aerosol (indicated by the wind field analysis in Figure 2).

The monsoon progression influence on aerosol chemical and physical properties can also be probed through analysis of ratios of various aerosol and gaseous species. In Figure 9 the BC:CO ($\mu gm^{-3}ppbv^{-1}$) and OA:BC increment ratios are presented. The ratio of BC:CO provides information on aerosol washout occurring as the monsoon progresses, and provides

details surrounding the primary aerosol present. As the monsoon progressed over northern India, the BC:CO ratio decreases especially in the IGP, indicative of wet removal. High ratios of OA:BC indicate that larger amounts of secondary organic aerosol (SOA) are being formed, whereas low OA:BC ratios are more often associated with a greater contribution from anthropogenic sources. The OA:BC ratio over northern India increased with the progression of the monsoon, which is likely be due to an increase in biogenic emissions.

**3.2 Vertical distribution of aerosol during the monsoon progression**

Vertical profiles were also carried out using the aircraft in NW India (Jaipur (26.91°N 75.79°E)/Jodhpur (26.24°N 73.02°E)), the IGP (Lucknow (26.85°N 80.95°E)) and NE India (Bhubaneswar (20.30°N 85.83°E)) during both the pre-monsoon (Sect. 3.2.1) and monsoon (Sect. 3.2.2) seasons.



### 3.2.1 Pre-monsoon

Inside the IGP, it has been shown that organic aerosol dominated in the boundary layer (as seen in Figure 10) however there was significant structure in the vertical. During the pre-monsoon, there was a clear thermodynamic separation in the column between the boundary layer and an elevated aerosol layer (EAL) aloft, as shown by the potential temperature (ө)

structure in Figure 3. The boundary layer is associated with a constant potential temperature with height, at the top of which increasing values of ө characterise the top of the convective mixing layer and marks the planetary boundary layer (PBL) height; above this is the EAL. These thermodynamic criteria were chosen due to use in previous literature (Seibert et al., 2000; Eresmaa et al., 2006; Haeffelin et al., 2012).

Inside the IGP, organic aerosol dominated the aerosol loading (43%) up to ~2 km with mass concentrations of ~15

$\mu g/m^3$ and is consistent with the boundary layer transects already discussed. Black carbon mass concentrations were largest in the boundary layer in the IGP (~2 $\mu g/m^3$). Other inorganic aerosol species also had significant mass concentrations in the boundary layer, with sulphate, ammonium and nitrate each contributing ~5$\mu g/m^3$. Above the boundary layer, an EAL existed between 2 and 7 km with sulphate the dominating aerosol species (50%) at concentrations ~10 $\mu g/m^3$. Other aerosol species were also seen in the EAL, but in much lower mass concentrations compared to within the boundary layer (5 $\mu g/m^3$ OA, 0.5

$\mu g/m^3$ BC).

Outside the IGP, the aerosol column had a different structure. In the boundary layer, the dominant aerosol species were sulphate (44%) up to around 1.5 km with mass concentrations of 8-10 $\mu g/m^3$. Black carbon mass concentrations were much lower outside the IGP (0.5 $\mu g/m^3$). Organic aerosol had significant mass concentrations in the boundary layer column but much lower than inside the IGP (3-5 $\mu g/m^3$ in the NW, and 1-2 $\mu g/m^3$ in the NE). It was similar for ammonium mass

concentrations also (2-3 $\mu g/m^3$) but even lower mass concentrations of nitrate in the column outside the IGP. The aerosol in the boundary layer extended vertically to a maximum height of 6-7 km, with an EAL present between 2-5 km dependent on location. For NE India near Bhubaneswar, the EAL was present between 2-4 km, with sulphate aerosol dominating similar to that over the IGP. Over NW India around Jaipur, the EAL was present between 3-6 km, again with sulphate dominating in this layer.

The aerosol physical properties throughout the column are shown in Figures 11 and 12. Inside the IGP, the aerosol in the boundary layer included both accumulation and coarse mode particles with scattering coefficient values of ~120 $Mm^{-1}$. Above the boundary layer in the EAL, the aerosol included both accumulation and coarse mode particles, with lower scattering coefficient values of ~80 $Mm^{-1}$. Outside the IGP, for the NW India region, a larger proportion of particles were in the coarse mode and scattering aerosol coefficients were consistent at 40-60 $Mm^{-1}$ through the entire atmospheric column, indicative of

sulphate and dust aerosol presence throughout.



### 3.2.2 Monsoon

The monsoon arrival had a strong impact on aerosol properties throughout the atmospheric column (as seen in Figures 10-12). During the pre-monsoon, the EAL was present across northern India, however this layer diminished as the monsoon continued. The monsoon system arrived first across the study region in NE India over Bhubaneswar, where large decreases in

total aerosol mass concentration were witnessed from 25-30 µg/m$^3$ to ~15 µg/m$^3$ in the EAL. Decreases occurred in the height at which the aerosol load was maximum, decreasing from 5 km in the pre-monsoon to below 2 km when the monsoon progressed over the region, with the EAL between 2-4 km being removed. These changes were also witnessed when the monsoon progressed and developed over the IGP and NW India locations. Inside the IGP, the EAL diminished with monsoon leaving behind aerosol only within the boundary layer, with mass concentrations decreased from ~30 µg/m$^3$ to ~20 µg/m$^3$. The

height of the maximum aerosol load decreased from 6 km to less than 2 km when the monsoon had fully covered this region. The changes in NW India were slightly different to the IGP and NE India however. This was in part due to the late arrival of the monsoon at these locations with the monsoon arrival 2 weeks late over LKN, so the flights that took place in this study were not carried out late enough into July 2016 to witness full monsoon development in NW India. Due to this, the changes to the aerosol profile were somewhat less severe. Aerosol mass concentrations throughout the profile decreased, but the maximum

aerosol height extent did not decrease as much as over the IGP and NE India (max height from 6 km to 5 km). The EAL over NW India did remain in position as the monsoon system progressed over the region, however decreases in mass concentration did occur. Over NW India it is clear that monsoon washout has been less significant due to the later onset date, which is reflected in the potential temperature and RH values indicating only partial monsoon arrival.

These changes in vertical structure were also reflected in the aerosol physical properties (as shown in Figures 11 and

12). Data coverage was poor with regards to capturing physical characteristics vertically as the monsoon progressed, however limited information is available across northern India. Despite this it is evident that aerosol number concentrations were decreasing, with largest decrease occurring for larger particle sizes, as shown by the number and volume size distributions. Over the IGP, coarse mode aerosol was dominant in the pre-monsoon but as the monsoon progressed there was a clear increase in the aerosol fine mode fraction alongside a decrease in the aerosol scattering coefficient to between 80-100 Mm$^{-1}$ in the

boundary layer and 20-40 Mm$^{-1}$ aloft (fine mode fraction is 1 when concentrations are very low). In addition, the height at which the coarse mode aerosol was present in abundance decreased as the monsoon progressed. Outside the IGP in NE India, similar trends in scattering coefficients were present from ~80 Mm$^{-1}$ in the pre-monsoon to ~50 Mm$^{-1}$ in the monsoon season. The impact in NW India was less pronounced due to the late monsoon onset in this region, however decreases in the maximum height of scattering aerosol were seen as the monsoon progressed.

### 4 Discussion and Conclusion

This study presents data for the first time from an aircraft platform collecting aerosol chemical composition and physical properties across northern India, thus providing a unique and unprecedented dataset. There is a clear need to conduct





measurements of the aerosol haze around the IGP, with the chemical and physical properties of the aerosol representing a significant unknown. The level of performance of the instruments provided high temporal and spatial resolution across northern India as the monsoon progressed, with the results in this study building upon previous work conducted over previous years and delivering new observations to meet some of the key measurement gaps.

In the vertical profile, inside the IGP, there was an elevated aerosol layer (EAL) present in the pre-monsoon which diminished as the monsoon progressed. With organic matter and fine mode absorbing aerosol dominating in the boundary layer inside the IGP, the EAL was dominated by $SO_4$ and coarse mode aerosol with lower mass concentrations of other aerosol species. In contrast, outside the IGP, the vertical profile was dominated by $SO_4$ and coarse mode aerosol in and above the boundary layer. The data presented in this study fills significant gaps in previous understanding, due to the temporal and spatial

coverage of the aircraft platform. Column and vertical profile measurements have been carried out in our study across northern India, where, to date, intensive measurements are lacking as they have only been carried out over the Arabian Sea or Bay of Bengal during the winter or earlier in the pre-monsoon season (Padmakumari et al., 2013). Sarangi et al. (2016) explain that the EAL over northern India consists of fine-mode anthropogenic aerosol alongside coarse-mode natural aerosol, i.e. dust, but the exact physical and chemical composition was unknown. Previous work in March and April has shown elevated aerosol

layers extended above the boundary layer to an altitude of ~5km (Kulkarni et al., 2012; Babu et al., 2016; Sarangi et al., 2016), but our in-situ measurements were able to quantify the aerosol present and its characteristics above these heights during the pre-monsoon and how they developed as the monsoon progressed. Maximum black carbon mass concentrations of ~0.5-1µg/m$^3$ occurred at an altitude of ~2 km in previous studies (Moorthy et al., 2009; Nair et al.; 2013), consistent with the EAL in our data present between 2-7 km, with similar concentrations of ~0.5 µg/m$^3$ of BC in the elevated layers. Our study also

highlights how the monsoon progresses over northern India and the impact of this on the EAL, with the diminishing mass concentrations and maximum height of aerosol particles. Throughout both the boundary layer and aloft, low $NO_3$ mass concentrations were present during the pre-monsoon (0-1 µg/m$^3$) with increases as the monsoon progressed (~5 µg/m$^3$). Coupled with large $SO_4$ mass concentrations, Sharma et al. (2014) explain that this pattern could be indicative of aerosols undergoing long-range transport.

By making use of in-situ and remote sensing techniques, studies have shown that dust aerosols are significant contributors to elevated aerosol loadings over the IGP during the pre-monsoon season (Gautam et al., 2010; Vaishya et al., 2018), along with input from absorbing aerosol species such as BC. The elevated aerosol layers, such as those in our study, have been highlighted to originate from Arabian and Thar Desert regions, driven by winds across the IGP around the height of 850 hPa/3-5 km (Das et al., 2013). The in-situ measurements however of aerosol optical properties have been lacking until

recently (Vaishya et al., 2018). The scattering coefficient values from our study provide useful comparison to those from Vaishya et al. (2018) despite the lack of in-situ absorption coefficient values through the vertical profile. In both datasets, vertical heterogeneity is clear in optical properties across northern India. Like Vaishya et al. (2018) the vertical variations are weak over NW India over locations such as Jaipur and Jodhpur but are much stronger over the central IGP (Lucknow) and NE India (Bhubaneswar) locations. Over NW India, scattering coefficient values are high in the boundary layer and aloft at values





comparable to Vaishya et al. (2018) around 50-60 Mm$^{-1}$. In the IGP around Lucknow, in a similar region to the central IGP location of Varanasi in Vaishya et al. (2018), the scattering coefficients are high in the boundary layer (80-120 Mm$^{-1}$), with decreased values in the elevated aerosol layer (~60 Mm$^{-1}$). Finally, in NE India, the scattering coefficient values are consistent and lower than the NW and IGP profiles, with coefficients of 60-70 Mm$^{-1}$ in the boundary layer and very low aloft. These

findings are consistent with the others in our study, highlighting the presence of larger, dust aerosol particles aloft with varying properties in the boundary layer dependent on location across northern India. Our scattering coefficient values also show agreement with previous literature values, such as Babu et al. (2016) for all locations, and Ram et al. (2016) for Lucknow.

      The spatial distribution of sub-micron aerosol chemical composition and physical properties has been characterised here based upon airborne measurements throughout the vertical column across northern India, during the pre-monsoon and

monsoon seasons. Aerosol source analysis in the literature can provide useful context for the aerosol composition presented. Across the IGP, it has been found that the residential sector provides the greatest particulate emissions, over double the emissions from large industry and transport (Pandey et al., 2014). Organic matter (wood burning stoves and open fires) and black carbon aerosol (kerosene lamps, woodstoves and agricultural residue-based stoves) are known to arise largely from residential fuel burning activities (Fleming et al., 2018), consistent with our findings as these practices are seen widely across

the IGP region where OM and BC are greatest. Distinctive to the IGP were high OA:BC ratios (10-15), which are somewhat higher than for similar polluted environments such as China ratios of ~4 (Ho et al., 2006). It has been indicated that OA:BC ratios over India can be greater than expected due to the dominance of biomass burning rather than fossil fuel emissions, with highest values in the pre-monsoon season (Ram and Sarin, 2011; Bisht et al., 2015; Kumar and Yadav, 2016) as seen in our data. Outside the IGP, the aerosol species concentrations are consistent with the aerosol sources, with more informal industries

and manufacturing industries present that form high concentrations of SO$_4$ aerosol through the conversion of sulphur dioxide (SO$_2$) into sulphuric acid (H$_2$SO$_4$). Ammonium mass concentrations correlate well with SO$_4$ regional patterns within the boundary layer, which is a strong indication of (NH$_4$)$_2$SO$_4$ formation in the atmosphere.

      This source information, coupled with the extensive in-situ measurements, will prove pivotal in producing more accurate climate models and being able to calculate the energy balance and climate forcing accurately through the monsoon.

**Competing interests**

The authors declare that they have no conflict of interest.

**Author contributions**

JB was responsible for the AMS and SP2 instrument operation in the field, data processing, data analysis, and the writing of this paper. JA and PW were responsible for the maintenance and running of the AMS prior to and during the campaign, and

DL supplied expertise and operation of the SP2 prior to and during the campaign. SK also contributed to the operation of the



AMS and SP2 in the field. JH, EH, SB, SS, AT and HC were the project investigators for this campaign. CF and JL were responsible for the data processing of the PCASP, CDP, Nephelometer and PSAP data. DS processed and analysed the lidar data.

## 5 Acknowledgements

5   We would like to thank those involved in the SWAAMI project, which is part of the larger MONSOON project. This includes the Facility for Airborne Atmospheric Measurements (FAAM) and AirTask who manage and operate the BAe-146 Atmospheric Research Aircraft, which is jointly funded by the Natural Environmental Research Council (NERC) and the Met Office. A number of institutions were involved in logistics, planning and support of the MONSOON campaign: The Met Office, University of Reading, Vikram Sarabhai Space Centre India, and the Indian Institute of Science India. ERA-Interim
10  wind field data was provided courtesy of ECMWF. The lead author was supported by a NERC studentship grant NE/L002469/1 and the work supported by NERC grant number NE/L013886/1.



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



**Figures**

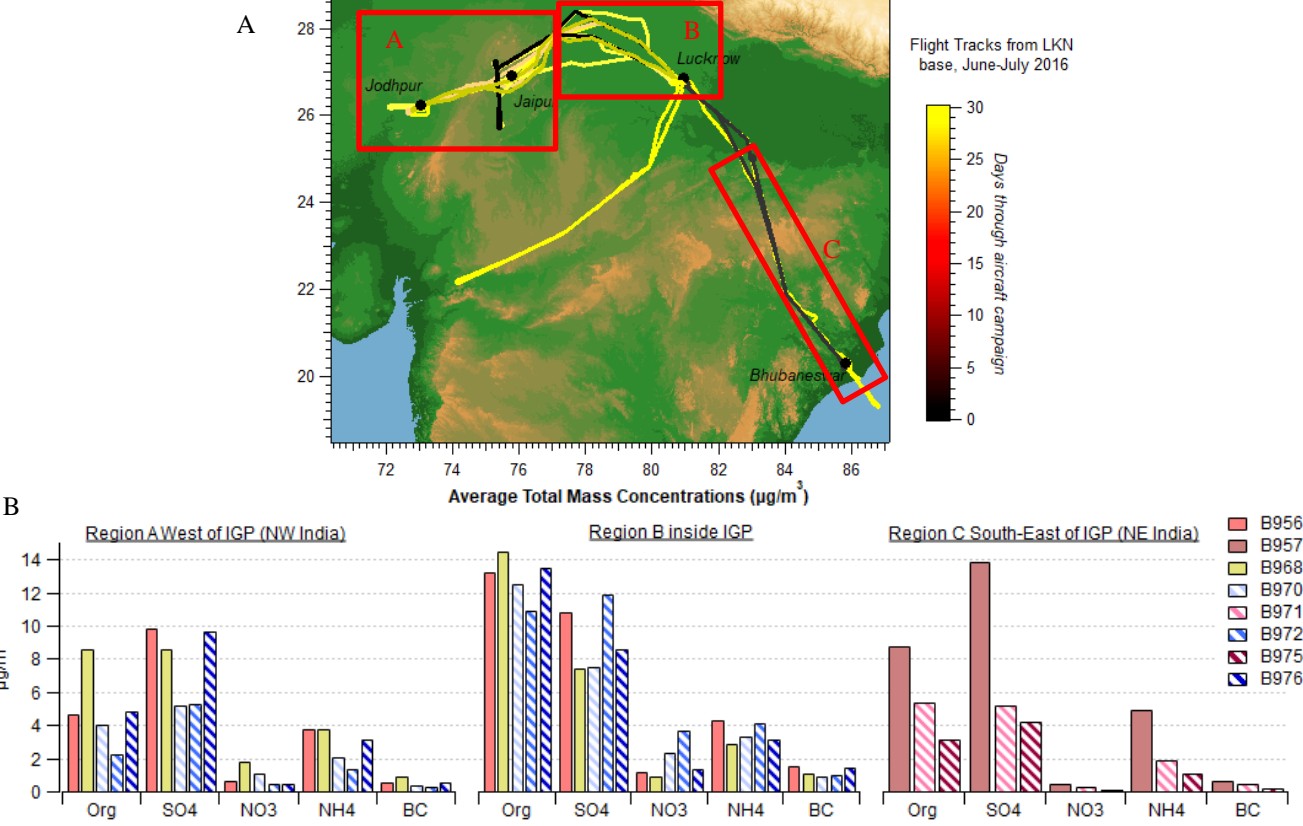

**Figure 1 Flight tracks of the BAe-146 aircraft for the campaign across India during the pre-monsoon and monsoon seasons of 2016. Panel A presents the flight paths considered by this analysis and are described in the main text and Table 1. Straight Level Run (SLR) boundary layer sections are split by region (A) West of IGP, (B) IGP, and (C) South-East of IGP. Panel B highlights a boundary layer data summary of the AMS and SP2 median mass concentrations (μg/m³) for each zone (A, B and C) identified across Northern India.**





| FLIGHT | SEASON | DATE | DEPART (Z) | RETURN (Z) | DURATION (hh:mm) | OPERATING REGION |
|--------|--------|------|-----------|-----------|------------------|------------------|
| B956 | PM | 11/06 | 03:05 | 07:36 | 04:31 | W |
| B957 | PM | 12/06 | 05:30 | 09:26 | 03:56 | E |
| B968 | PM/M | 30/06 | 03:32 | 07:28 | 03:56 | W |
| B970 | PM/M | 03/07 | 04:46 | 08:42 | 03:56 | W |
| B971 | PM/M | 04/07 | 05:40 | 10:05 | 04:25 | E |
| B972 | M | 05/07 | 03:27 | 07:29 | 04:02 | W |
| B973 | M | 06/07 | 02:10 | 06:41 | 04:31 | W |
| B974 | M | 07/07 | 04:27 | 08:18 | 03:51 | W |
| B975 | M | 09/07 | 04:29 | 09:04 | 04:35 | E |
| B976 | M | 10/7 | 04:23 | 08:51 | 04:28 | W |

**Table 1 Flight summary for operations included in this study. All flights were conducted in Northern India in the pre-monsoon (PM) and monsoon (M) season (PM/M season refers to transition period of when the monsoon was arriving in Northern India). The dates of the flights are shown, with their respective region of study.**





**Figure 2 Mean meteorological conditions in the boundary layer (950hPa pressure altitude) (panel A and C) and aloft (650hPa pressure altitude) (panel B and D) for the pre-monsoon and monsoon flights. The wind direction is displayed using white arrows on each of the plots, with temperature (Kelvin) and relative humidity (%) indicated with the colour scale. The data used in the maps is ERA-Interim (Dee et al., 2011). Panels E and F present 1-degree gridded precipitation (mm) data based on 2140 weather stations over India, again for the pre-monsoon and monsoon flights, from the Indian Meteorological Department (IMD).**





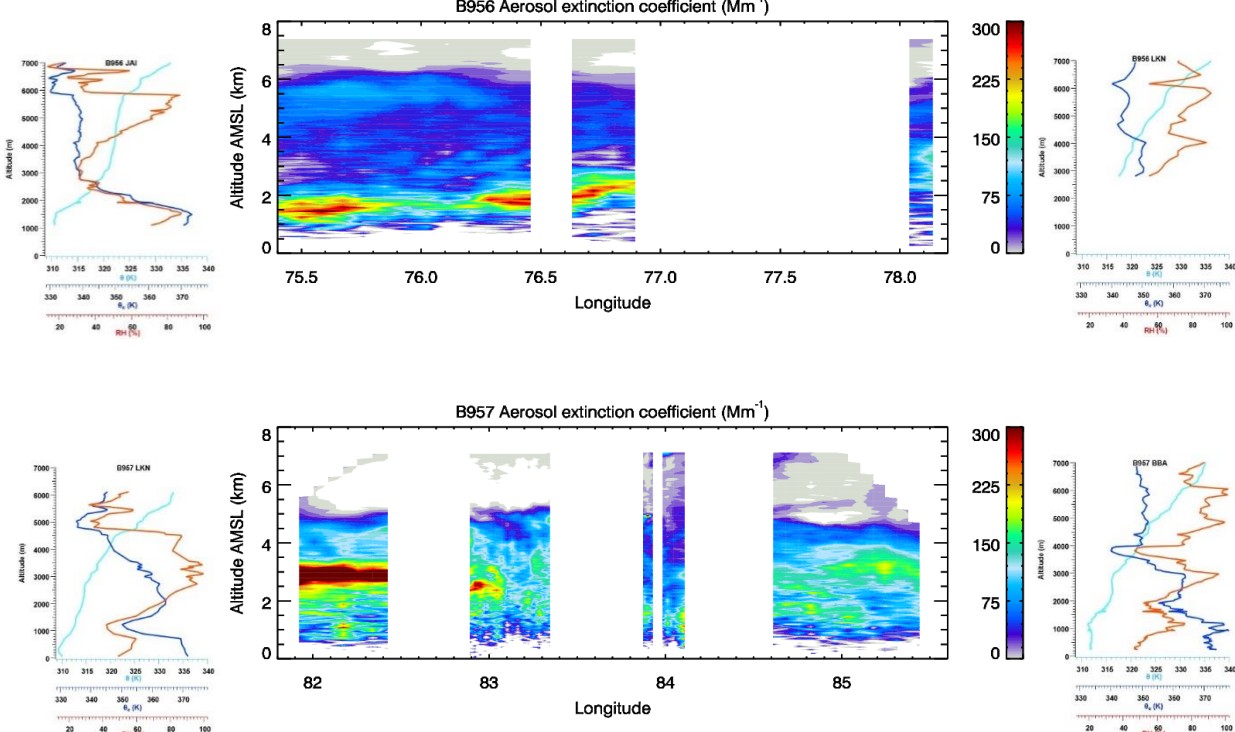

**Figure 3 Aerosol extinction coefficient estimated from backscatter lidar observations, for the indicated flights of B956 (11/06) and B957 (12/06) in the pre-monsoon season along the longitudinal scale 75ºE to 90ºE. Gaps indicate missing data, due to either rejection of whole profiles or rejection of the portion of profiles affected by clouds. The thermodynamic plots of potential temperature (θ), equivalent potential temperature (θₑ) and relative humidity on the right and left of the lidar profiles, are for the location indicated by title of plot (JAI=Jaipur (26.91ºN, 75.79ºE), LKN=Lucknow (26.85ºN, 80.95ºE), BBA=Bhubaneswar (20.30ºN, 25.83ºE)).**





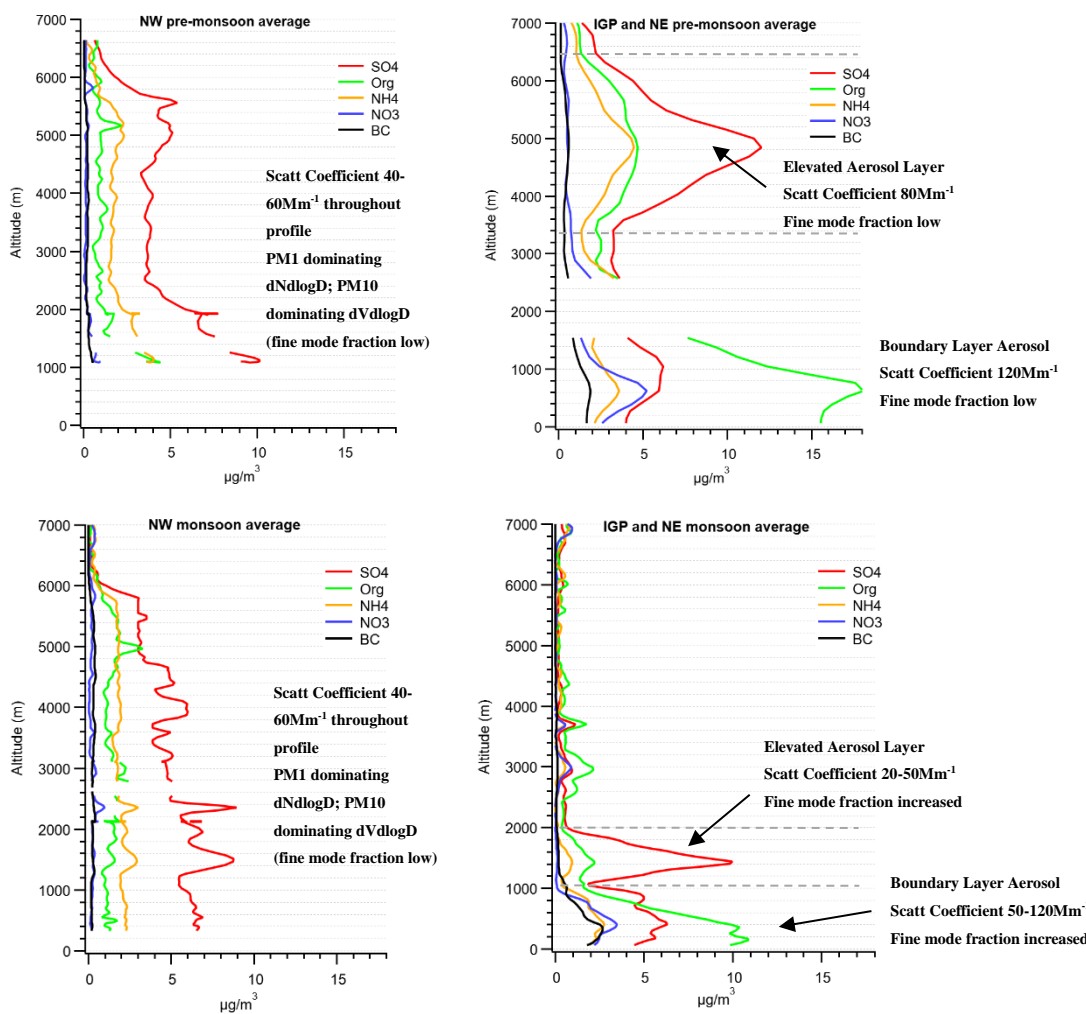

**Figure 4 Aerosol vertical profile summary for pre-monsoon and monsoon seasons during the campaign, for NW India and the IGP/NE India.**



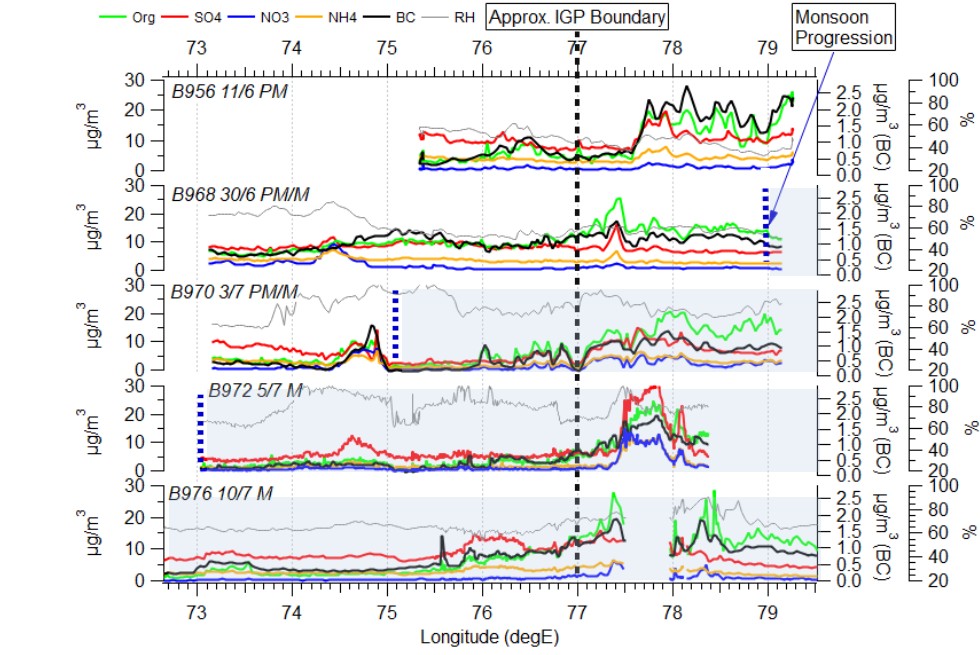

**Figure 5 Horizontal spatial aerosol chemical composition plots for pre-monsoon (PM) and monsoon (M) seasons in Northern India, from LKN to JAI/JOD. The plots highlight the IGP region (77ºE-eastwards) and outside the IGP (73-77ºE) separated by the vertical black dashed line. The monsoon progression is indicated by the blue shaded region.**

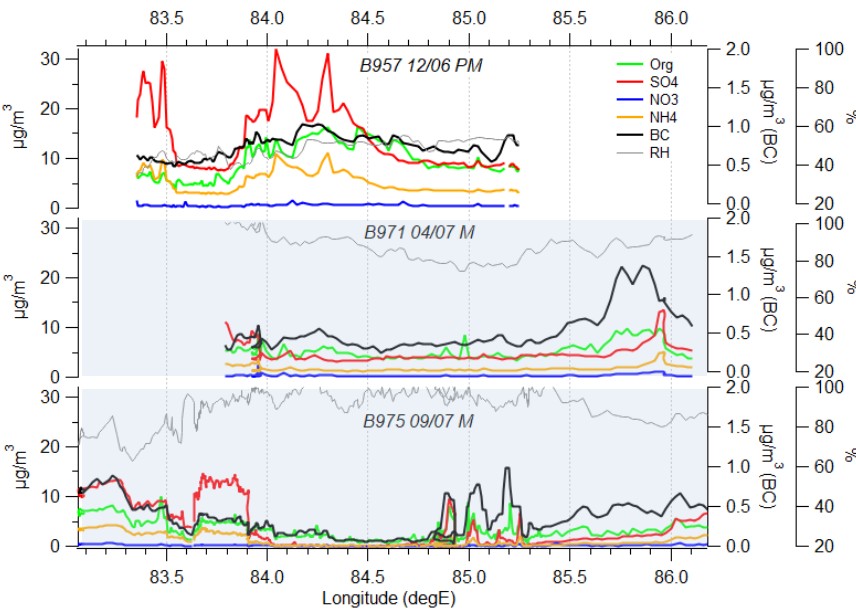

**Figure 6 Horizontal spatial aerosol chemical composition plots for PM and M seasons in Northern India, from LKN to BBA. The monsoon progression is indicated by the blue shaded region.**




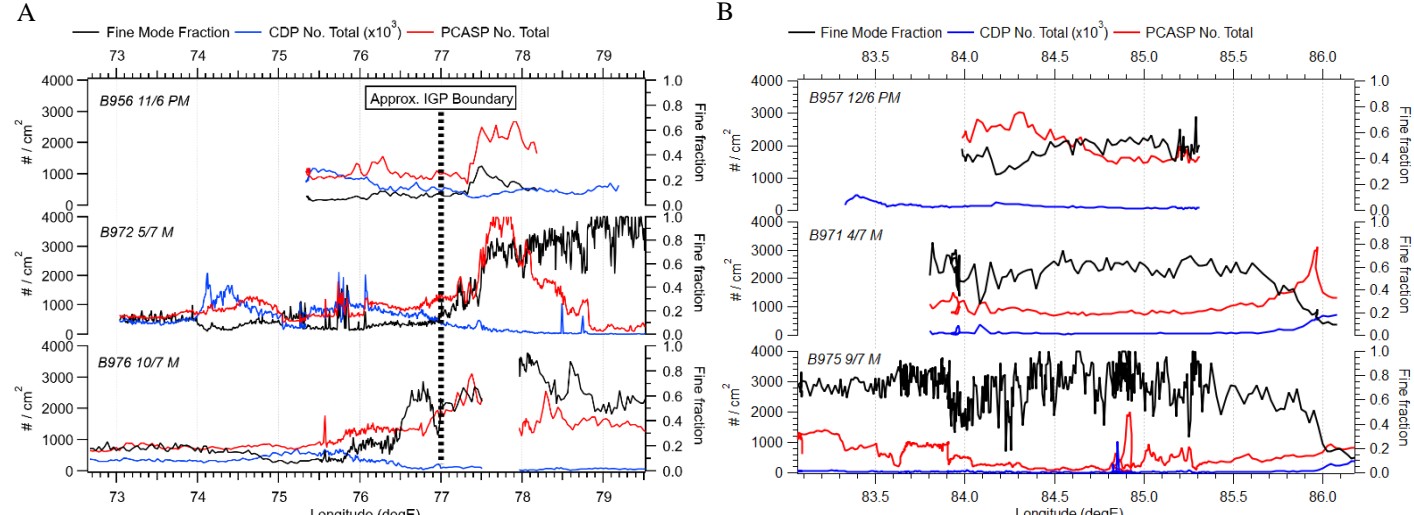

**Figure 7 Horizontal variability in aerosol physical properties (CDP no. total, PCASP no. total and Fine Mode fraction) for PM and M seasons in Northern India, from (A) LKN to JAI/JOD and (B) LKN to BBA.**





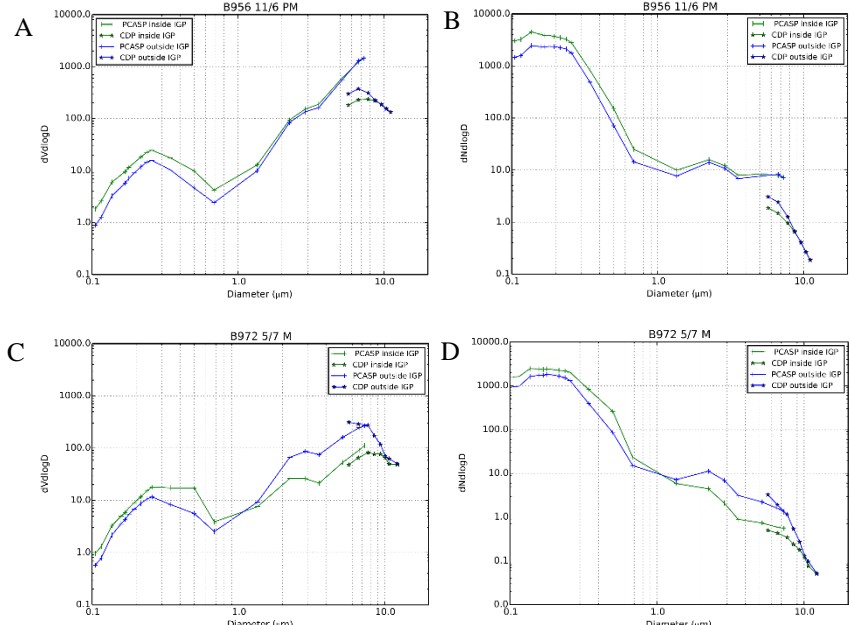

**Figure 8 Volume size (dVdlogD) and number size (dNdlogD) distribution plots from the CDP and PCASP data, for (A+B) B956 (pre-monsoon) and (C+D) B972 (monsoon) flights. The blue (green) data points are for outside the IGP (inside the IGP).**

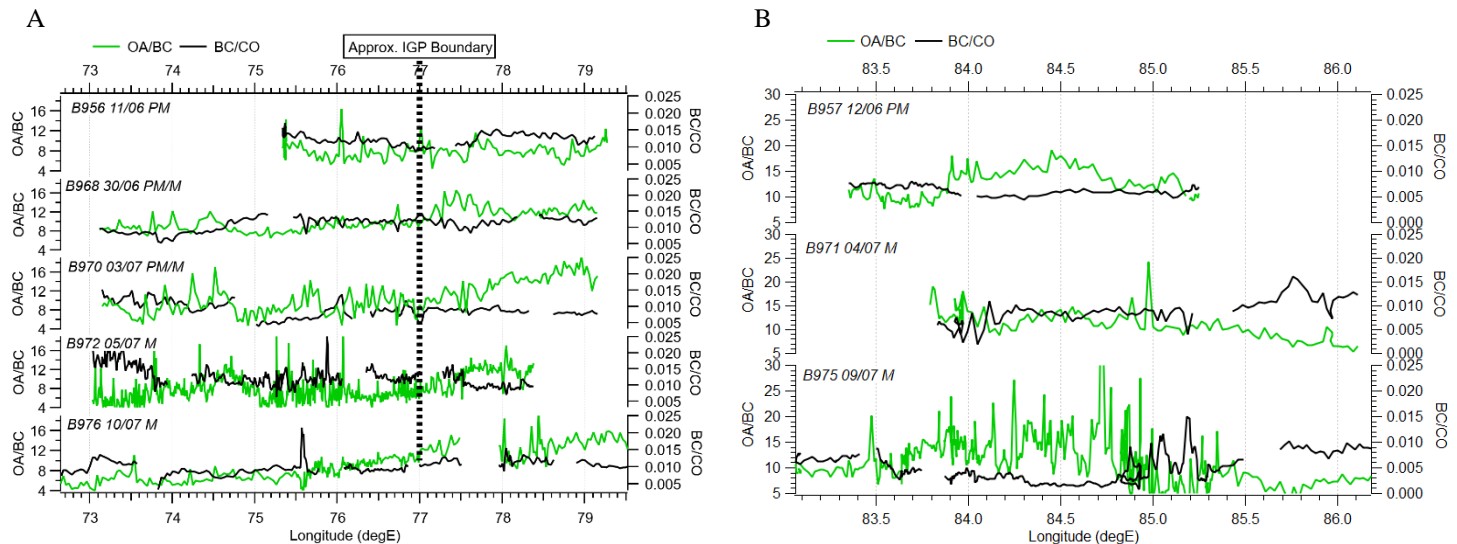

**Figure 9 Horizontal spatial aerosol chemical composition (Organics:Black Carbon (OA:BC) and Black Carbon:Carbon Monoxide (BC:CO) ratios) for PM and M seasons in Northern India, from (A) LKN to JAI/JOD and (B) from LKN to BBA.**





**Figure 10 Vertical profile aerosol chemical composition for (A) Jaipur/Jodhpur in NW India, (B) Lucknow in the IGP, and (C) Bhubaneswar in NE India. Data presented here are for the pre-monsoon (PM) and monsoon (M) season. The Figures show AMS (bottom x-axis) and SP2 (BC) (top x-axis).**



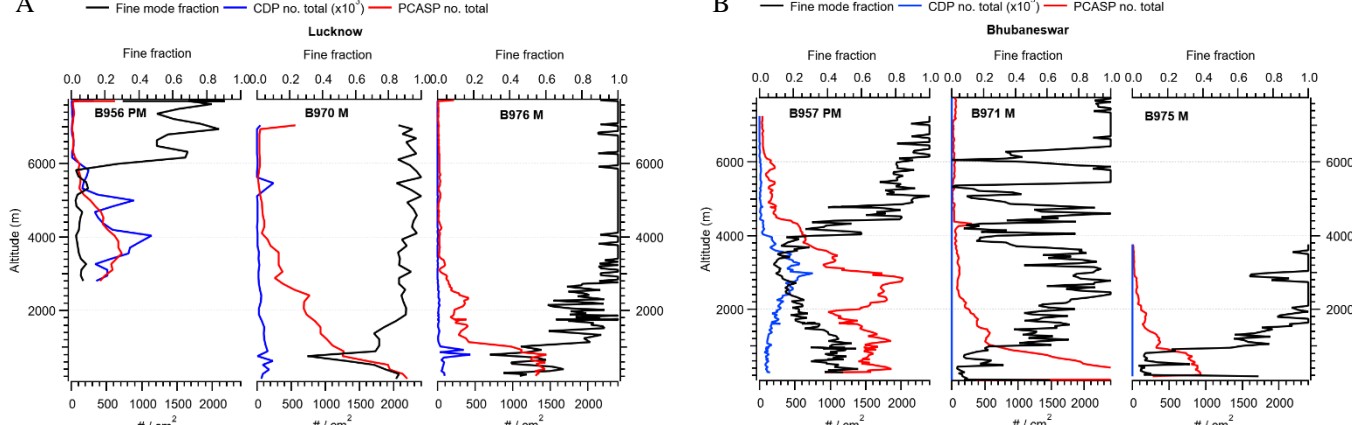

**Figure 11 Vertical Profiles of aerosol physical properties (CDP no. total, PCASP no. total and Fine Mode fraction) for (A) Lucknow and (B) Bhubaneswar for one PM and two M flights.**

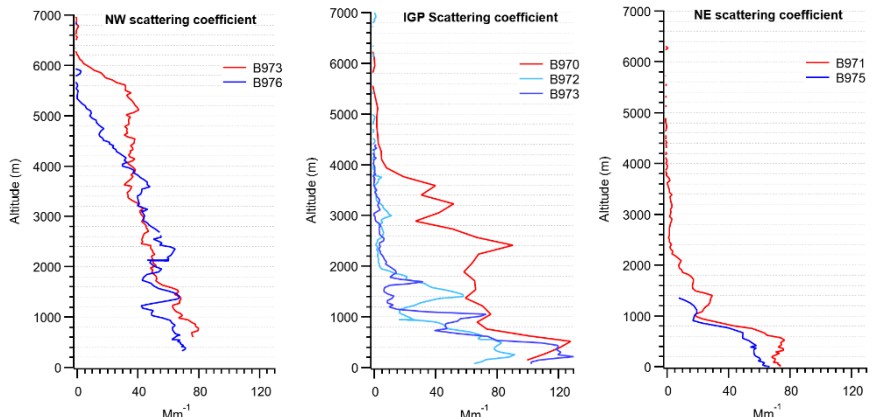

**Figure 12 Aerosol scattering coefficient (Mm⁻¹) in the vertical profile for NW India (Jaipur/Jodhpur 26.91ºN 75.79ºE), IGP (Lucknow 26.85ºN 80.95ºE) and NE India (Bhubaneswar 20.30ºN 85.83ºE). The colours represent different science flights, with various progression into the monsoon season.**





**Appendix**

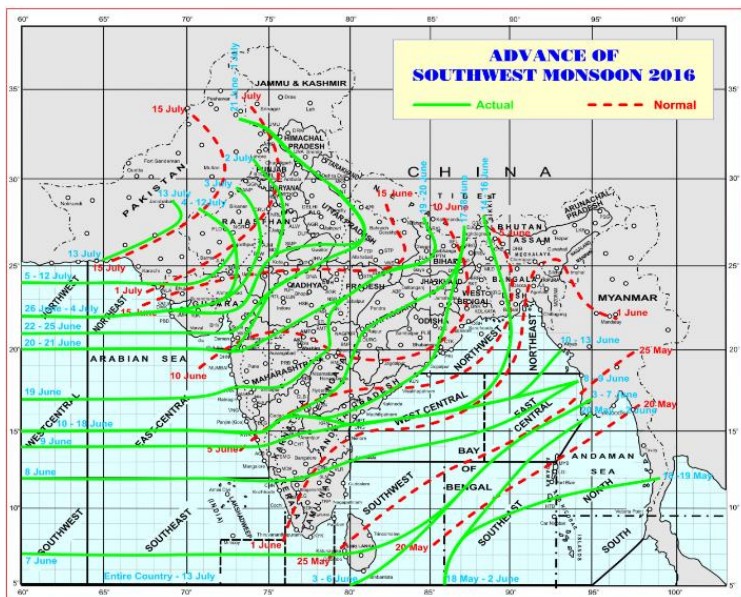

**Figure A1 Advance of the Indian summer monsoon of 2016 (India Meteorological Department, Ministry of Earth Sciences, India)**