# Peer review of "Vertical and horizontal distribution of sub-micron aerosol chemical composition and physical characteristics across Northern India, during the pre-monsoon and monsoon seasons"

_Atmospheric Chemistry and Physics, 2018_

## Referee Comment (RC1) · Anonymous Referee #1 · 18 Jan 2019

Manuscript # acp-2018-1109

Manuscript title: Vertical and horizontal distribution of sub-micron aerosol chemical composition and physical characteristics across Northern India, during the pre-monsoon and monsoon seasons

Authors: James Brooks et al.

Dear Editor/Authors,

The submitted manuscript presents detailed airborne in situ measurements of aerosols

taken during different flights over northern India covering pre-monsoon and monsoon seasons. The characteristics of aerosols over the region are presented regarding high-quality vertical and spatial measurements of optical, microphysical, and chemical composition of aerosols. The measurement dataset reveals higher concentration of organic matter followed by sulfate, ammonium, and black carbon mostly confined within the boundary layer inside the Indo-Gangetic Plain (IGP)–one of the most densely populated areas of the world. Above the boundary layer, the measurements show the dominance of coarse mode dust aerosols between 3-6 km transported from the adjacent Thar Desert. Outside the IGP, the sulfate component is found to dominate the aerosol mass followed by other species. Upon arrival of monsoon season and then onwards, the mass concentration of aerosols is found to decrease significantly, by $\sim$50% and $\sim$30%, outside and inside the IGP region, respectively.

The results presented in the paper bring an unprecedented set of information about aerosol spatial and vertical distribution, with its chemical analysis, over northern India, which can help constraint aerosol representation in the models and satellite-based remote sensing algorithms. However, first, it was a little surprise to me that authors didn't include the CALIOP space lidar data to complement and support (or not) their findings. CALIOP lidar provides a detailed vertical structure of aerosol backscatter and extinction that can be compared with the aircraft measurements for the consistency (or lack thereof) check. Second, the ground-based AERONET aerosol measurements at a couple of sites (Kanpur and Gandhi College) located in the center of IGP can also offer another perspective and correlation to the presented measurements. Authors are strongly recommended to add these two components to the article which, in my opinion, will further enhance the content and quality of the work.

Specific suggestions on the paper are listed below. The article is mostly well-written with some attention needed to improve the presentation, e.g., long sentences, punctuations. The content highlighted in the paper certainly fits into the scope of the ACP journal and can be published given that above two major concerns are addressed.

Thanks.

Specific comments:

It was a little surprise to me that authors didn't include the CALIOP space lidar data to complement and support (or not) their findings. CALIOP lidar provides a detailed vertical structure of aerosol backscatter and extinction that can be compared with the aircraft measurements for the consistency (or lack thereof) check. Second, the ground-based AERONET aerosol measurements at a couple of sites (Kanpur and Gandhi College) located in the center of IGP can also offer another perspective and correlation to the presented measurements. Authors are strongly recommended to add these two components to the article which, in my opinion, will further enhance the content and quality of the work.

CALIPSO browse images https://www-calipso.larc.nasa.gov/products/lidar/browse_images/production/

Daytime CALIOP/CALIPSO overpass on the Indian subcontinent on June 11th Night-time overpass on June 30th Daytime and nighttime overpass on July 11th

AERONET data over Kanpur and Gandhi College: https://aeronet.gsfc.nasa.gov/ AERONET volume size distribution and fine-mode fraction can be compared with air-craft measurements, at least on a qualitative sense.

Title: Remove 'comma' and 'the'

Abstract: Line 4: "...high mass concentration of dust(?)" Line 11: what is 'std'? Line 20-25: Elevated concentration of dust at altitudes >1.5 is a clear indication of dust transport from the Great Indian Desert, also called the Thar Desert, in northwestern India

Introduction Page 3, line 17: "...have been subject to analysis now for nearly two decades" Jethva et al. (2005) has been one of the early research works highlighted the seasonal variability of aerosols, both natural and anthropogenic, over the Indian sub-continent using satellite and ground measurements.

Jethva, H., S. K. Satheesh, and J. Srinivasan (2005), Seasonal variability of aerosols over the Indo-Gangetic basin, J. Geophys. Res., 110, D21204, doi:10.1029/2005JD005938.

Page 3, line 21-22: "Much uncertainty. . ..that determine the resultant climatic impact of aerosols as well as the regional air quality"

Methodology and Climatology Page 4, line 8: "A total of twenty-two science flights. . ."

Results: Page 8, line 14-15: Figure3: The extinction profile derived from lidar measurements show peak concentration between 1.5 to 2 km for June 11th-flight B956; that for June 12th-flight B957 shows centroid of the aerosol layer at 3 km with the presence of aerosols with reduced extinction from 6 km to all the way to near-surface. The author needs to reword the interpretation of Figure 3.

Last paragraph: It is striking to me that NW region shows a minor peak in SO4 between 4 to 6 km, but the peak is much clearer and more pronounced over IGP. There is no doubt, in my opinion, that elevated peak in concentration over IGP is a result of transported dust from NW, likely from the Thar Desert, but it is intriguing and a bit counter-intuitive that such peak isn't observed over NW!

Page 9, Figure 5: Please mention in the caption that the data belongs to the first 1000 meters of the atmosphere. Page 10, first paragraph: Bringing here the size distribution retrievals from AERONET over Kanpur and Gandhi College stations for the same dates or nearby dates is necessary here to complement and compare the aircraft observations.

Page 13, line 1: ". . .aerosol haze in and around the IGP" Page 13, line 16: "aerosol presence"

Page 14, line 16: "such as China with a ratio of"

---

## Referee Comment (RC2) · Anonymous Referee #2 · 12 Feb 2019

Review on 'Vertical and horizontal distribution of sub-micron aerosol chemical composition and physical characteristics across Northern India, during the pre-monsoon and monsoon seasons' by James Brooks et al., (ACP-2018-1109)

This manuscript presents the results on physical and chemical properties of elevated aerosol layer and their vertical and horizontal distribution within and outside of IGP region over India using UK Facility for Airborne Atmospheric Measurements Bae-16 research aircraft measurements. In general, results presented in this manuscript are unique which are first of its kind and authors made nice compilation of physical and

chemical characteristics of elevated aerosol layers (EAL) over north part of India. First of all, I should congratulate the authors for bringing out this study. Authors also brought out the differences in aerosol characteristics during pre-monsoon and monsoon both within and outside the IGP regions which will be very useful for further understanding the role of these EAL on the background atmosphere. As rightly pointed out in the summary, the information provided in this paper is very much useful in producing more accurate climate models by estimating the energy balance and for getting insights on the climate forcing accurately.

In general, paper is well written and will be interest to the researchers working in this field and very apt to publish in journals like ACP. However, there are few mistakes and sometimes interpretation is missing at some instances including literature survey which demands careful editing or re-writing. Below are the some of the issues which authors may take care in revising the manuscript. Authors are strongly encouraged to revise and re-submit this manuscript.

Below are my specific comments/suggestions for the potential solutions which authors may consider for future analysis.

Specific comments/suggestions:

1. Number of aircraft flights mentioned in the text (section 2) are 22, but only 4 dates are selected (in abstract) representing pre-monsoon (2 days) and monsoon season (2 days). However, from Table 1, it is clear that flights are operated on 10 days. It is not clear that why all the aircraft measurements are not considered in the present analysis.

2. In Figure 3, Aerosol Extinction (AE) is shown for the two days in pre-monsoon season. Do you have similar profiles in monsoon season? It will be interesting to see the presence of elevated aerosol layer in monsoon season as also shown by Sinha et al. (2013) and very recently by Venkat Ratnam et al. (2018). Latter study showed the presence of elevated aerosol during monsoon season and wet scavenging is clearly brought out.

[Figure]

3. Elevated aerosol layers over Asian region are formed either due to convective or long-range transport. If it is through convective transport, chemical composition near surface should match with that observed aloft. In this study, surface concentrations of the chemical composition of the aerosol are missing. Information from either earlier published literature or their own surface measurements will be useful in interpreting the role of convective and long-range transport. I am wondering whether surface concentrations are measured with same set of instruments onboard aircraft before starting of the aircraft measurements each time?

4. One of the conclusions that the dust and sulphate dominated aerosol layer aloft was removed upon monsoon arrival matches with the findings of Vernier et al. (2018) where they found Nitrate as dominant source in the UTLS region using zero pressure balloons from Hyderabad whereas sulphate near the surface. Some of these finds are useful in further interpreting the results.

5. Are the profiles in figure 4 is the average of two flights in each season? If yes, it is better to show them separately to feel how different these profiles are within the same season (one can add as supplementary figure, if you feel that already figures are more).

6. Page 11, line 7: It is mentioned that elevated aerosol layer diminish as monsoon arrives leaving aerosol only within boundary layer. If monsoon washout (wet scavenging) is expected to remove the elevated aerosol layer, then throughout the profile including boundary layer aerosol should have also washed out? Why only elevated aerosol layer is diminished? Further there was no mention of rainout process which is also important during monsoon.

7. There was no mention of true boundary layer altitude though many times it is used. 950 hPa do not represent throughout the Boundary layer though it is expected that within boundary layer all the spices are well mixed. In figure 3 description, boundary layer is considered as 2-4 km?

8. As also pointed out by other reviewer, use of CALIPSO profiles (aerosol, extinction and aerosol types) will be very useful for two reasons. One is to know how good satellite retrievals over this region and also whether aerosol types obtained from CALIPSO match with aircraft observations or not.

Minor comments/suggestions:

There is repeation in the abstract (lines 17 and 24) ...that total mass reduced to 50%

Page 8 line 6: How average total mass concentrations are obtained is not clear.

Page 13, line 16: Quoted reference over China is almost a decade back where level of pollution over India may also be less at that time. Re-wording of this statement is required.

Additional References:

Sinha, P. R., Manchanda, R.K., Kaskaoutis, D.G., Kumar, Y.B., and Sreenivasan, S. (2013). Seasonal variation of surface and vertical profile of aerosol properties over a tropical urban station Hyderabad, India. J. Geophys. Res. 118, doi:10.1029/2012JD018039.

Vernier, J., et al. (2018). BATAL: The Balloon measurement campaigns of the Asian Tropopause Aerosol Layer. Bull. Amer. Meteor. Soc., https://doi.org/10.1175/BAMS-D-17-0014.1.

Venkat Ratnam, M., P. Prasad, M. Roja Raman, V. Ravikirana, S.V.B. Rao, B.V. Krishna Murthy and A. Jayaraman. (2018). Role of dynamics on the formation and maintenance of the elevated aerosol layer during monsoon season over south-east peninsular India. Atmospheric Environment. 188. 10.1016/j.atmosenv.2018.06.023.

———————————————

---

## Author Comment (AC1) · 18 Mar 2019

**We thank both reviewers for their constructive comments, which as outlined below have helped improve the manuscript. This document outlines the review comments in plain italics, followed by the authors replies in bold.**

**acp-2018-1109-RC1**

*The submitted manuscript presents detailed airborne in situ measurements of aerosols taken during different flights over northern India covering pre-monsoon and monsoon seasons. The characteristics of aerosols over the region are presented regarding high quality vertical and spatial measurements of optical, microphysical, and chemical composition of aerosols. The measurement dataset reveals higher concentration of organic matter followed by sulfate, ammonium, and black carbon mostly confined within the boundary layer inside the Indo-Gangetic Plain (IGP)–one of the most densely populated areas of the world. Above the boundary layer, the measurements show the dominance of coarse mode dust aerosols between 3-6 km transported from the adjacent Thar Desert. Outside the IGP, the sulfate component is found to dominate the aerosol mass followed by other species. Upon arrival of monsoon season and then onwards, the mass concentration of aerosols is found to decrease significantly, by ∼50% and ∼30%, outside and inside the IGP region, respectively. The results presented in the paper bring an unprecedented set of information about aerosol spatial and vertical distribution, with its chemical analysis, over northern India, which can help constraint aerosol representation in the models and satellite-based remote sensing algorithms. However, first, it was a little surprise to me that authors didn't include the CALIOP space lidar data to complement and support (or not) their findings. CALIOP lidar provides a detailed vertical structure of aerosol backscatter and extinction that can be compared with the aircraft measurements for the consistency (or lack thereof) check. Second, the ground-based AERONET aerosol measurements at a couple of sites (Kanpur and Gandhi College) located in the center of IGP can also offer another perspective and correlation to the presented measurements. Authors are strongly recommended to add these two components to the article which, in my opinion, will further enhance the content and quality of the work. Specific suggestions on the paper are listed below. The article is mostly well-written with some attention needed to improve the presentation, e.g., long sentences, punctuations. The content highlighted in the paper certainly fits into the scope of the ACP journal and can be published given that above two major concerns are addressed.*

**We thank the Reviewer for their encouraging view of our work and the useful suggestions for consideration of CALIPSO remote-sensing and AERONET ground-based data, which we have incorporated in the revised manuscript.**

**SPECIFIC COMMENTS**

*It was a little surprise to me that authors didn't include the CALIOP space lidar data to complement and support (or not) their findings. CALIOP lidar provides a detailed vertical structure of aerosol backscatter and extinction that can be compared with the aircraft measurements for the consistency (or lack thereof) check. Second, the groundbased AERONET aerosol measurements at a couple of sites (Kanpur and Gandhi College) located in the center of IGP can also offer another perspective and correlation to the presented measurements. Authors are strongly recommended to add these two components to the article which, in my opinion, will further enhance the content and quality of the work.*

*CALIPSO browse images https://www-calipso.larc.nasa.gov/products/lidar/browse_images/production/ Daytime CALIOP/CALIPSO overpass on the Indian subcontinent on June 11th Nighttime overpass on June 30th Daytime and nighttime overpass on July 11th*

*AERONET data over Kanpur and Gandhi College: https://aeronet.gsfc.nasa.gov/ AERONET volume size distribution and fine-mode fraction can be compared with aircraft measurements, at least on a qualitative sense.*

**This information has now been added to the paper. The CALIPSO satellite retrieval from the daytime pass on June 11th has been added and is Figure 4. The CALIPSO data shows useful comparison and agreement with the on-board lidar plot in Figure 3. The aerosol structure is comparable between both lidar plots, with CALIPSO successfully capturing the maximum aerosol height during the pre-monsoon in northern India (~6-7 km) along with the values shown representing dust aerosol dominance, consistent with the literature for the pre-monsoon season (page 8 line 20-24). While it does not show the distinct aerosol layers that the AMS and SP2 data reveal, it does reinforce the features shown from the on-board lidar in Figure 3.**

*Title: Remove 'comma' and 'the'*

**Complied with, we agree that this is better for the title.**

*Abstract: Line 4: ". . .high mass concentration of dust(?)" Line 11: what is 'std'?*

**"std" is referring to a correction to the aerosol mass concentrations, to present them at standard temperature (273.15 K) and pressure (1013.25 hPa) conditions. This is explained for example in section 2.2 (page 6 line 24). To avoid any confusion, as the correction is explained in the main text body, we have decided to remove the 'std' from the abstract.**

*Line 20-25: Elevated concentration of dust at altitudes >1.5 is a clear indication of dust transport from the Great Indian Desert, also called the Thar Desert, in northwestern India*

**This point has been noted and clarified in the text, thank you.**

*Introduction Page 3, line 17: ". . .have been subject to analysis now for nearly two decades" Jethva et al. (2005) has been one of the early research works highlighted the seasonal variability of aerosols, both natural and anthropogenic, over the Indian sub-continent using satellite and ground measurements. Jethva, H., S. K. Satheesh, and J. Srinivasan (2005), Seasonal variability of aerosols over the Indoâ˘*
10 *RGangetic basin, J. Geophys. Res., 110, D21204, ˘ doi:10.1029/2005JD005938.*

**Noted – we have included this useful reference into the introduction.**

*Page 3, line 21-22: "Much uncertainty. . ..that determine the resultant climatic impact of aerosols as well*
15 *as the regional air quality"*

**Complied with, the text has been altered.**

*Methodology and Climatology Page 4, line 8: "A total of twenty-two science flights. . ."*

**Complied with and clarified this point in the text. As explained in the response to referee 2, this text has been altered to 10 flights as this study only involves 10 science flights from the campaign, not the 22 that were flown overall for the project. The other flights were in regions not applicable to this paper.**

*Results: Page 8, line 14-15: Figure3: The extinction profile derived from lidar measurements show peak concentration between 1.5 to 2 km for June 11th-flight B956; that for June 12th-flight B957 shows centroid of the aerosol layer at 3 km with the presence of aerosols with reduced extinction from 6 km to all the way to near-surface. The author needs to reword the interpretation of Figure 3.*

**The text has been adapted to clarify the features shown in the lidar extinction profile, in order to accurately interpret Figure 3 (page 8 line 14-16).**

*Last paragraph: It is striking to me that NW region shows a minor peak in SO4 between 4 to 6 km, but*
35 *the peak is much clearer and more pronounced over IGP. There is no doubt, in my opinion, that elevated peak in concentration over IGP is a result of transported dust from NW, likely from the Thar Desert, but it is intriguing and a bit counter-intuitive that such peak isn't observed over NW!*

**Firstly we thank the referee for this insight into our data. In our study, we do not try and examine exactly where the sulphate aerosol is coming from. The regional air mass directions are consistently from the northwest during the pre-monsoon, but it is difficult to analyse the sources of the sulphate aerosol. Oil production and power plants are seen to be present in northwest India and the surrounding areas (Garg and Shukla, 2009) which could be producing the sulphate, but given the heterogeneous nature of the wind patterns there would be very large differences in the sulphate mass concentrations with only a small change in the air mass direction. However, we understand this is an area that needs considering, and we have now added this awareness of the possibility of sulphate sources in the text (section 4; page 13 line 27-31).**

**Garg, A. and Shukla, P.R.: Coal and energy security for India: Role of carbon dioxide (CO2) capture and storage (CCS). *Energy*, *34*(8), pp.1032-1041, 2009.**

*Page 9, Figure 5: Please mention in the caption that the data belongs to the first 1000 meters of the atmosphere.*

**Complied with – this is now mentioned in the figure caption.**

*Page 10, first paragraph: Bringing here the size distribution retrievals from AERONET over Kanpur and Gandhi College stations for the same dates or nearby dates is necessary here to complement and compare the aircraft observations.*

**Complied with – the AERONET size distribution over Kanpur is included, however not Ghandi College due to lack of data during the measuring period for the study. The data for Kanpur does show agreement which complements the aircraft observations, with the reduction in volume size distribution especially in the larger aerosol size range as the monsoon arrives. The AERONET volume size distributions have been added to Figure 9 (page 29).**

*Page 13, line 1: ". . .aerosol haze in and around the IGP" Page 13, line 16: "aerosol presence"*

**This is now complied with thank you – edits have been made in the text.**

*Page 14, line 16: "such as China with a ratio of"*

**Noted thank you; we have adapted the text as such**

**acp-2018-1109-RC2**

*This manuscript presents the results on physical and chemical properties of elevated aerosol layer and their vertical and horizontal distribution within and outside of IGP region over India using UK Facility for Airborne Atmospheric Measurements Bae-16 research aircraft measurements. In general, results*
5 *presented in this manuscript are unique which are first of its kind and authors made nice compilation of physical and chemical characteristics of elevated aerosol layers (EAL) over north part of India. First of all, I should congratulate the authors for bringing out this study. Authors also brought out the differences in aerosol characteristics during pre-monsoon and monsoon both within and outside the IGP regions which will be very useful for further understanding the role of these EAL on the background atmosphere.*
10 *As rightly pointed out in the summary, the information provided in this paper is very much useful in producing more accurate climate models by estimating the energy balance and for getting insights on the climate forcing accurately. In general, paper is well written and will be interest to the researchers working in this field and very apt to publish in journals like ACP. However, there are few mistakes and sometimes interpretation is missing at some instances including literature survey which demands careful*
15 *editing or re-writing. Below are the some of the issues which authors may take care in revising the manuscript. Authors are strongly encouraged to revise and re-submit this manuscript. Below are my specific comments/suggestions for the potential solutions which authors may consider for future analysis.*

**We thank the Reviewer for their encouraging comments on our work and the suggestions made for**
20 **improvement.**

**SPECIFIC COMMENTS**

*1. Number of aircraft flights mentioned in the text (section 2) are 22, but only 4 dates are selected (in*
25 *abstract) representing pre-monsoon (2 days) and monsoon season (2 days). However, from Table 1, it is clear that flights are operated on 10 days. It is not clear that why all the aircraft measurements are not considered in the present analysis.*

**Noted – we acknowledge there was clarity required for the flight numbers during the study. There**
30 **was also a southern India flying period, that brings the total flight number to 22, but as these are not included in the scope of this study we have edited the text to make it clear that 10 flights are considered in this study.**

*2. In Figure 3, Aerosol Extinction (AE) is shown for the two days in pre-monsoon season. Do you have*
35 *similar profiles in monsoon season? It will be interesting to see the presence of elevated aerosol layer in monsoon season as also shown by Sinha et al. (2013) and very recently by Venkat Ratnam et al. (2018).*

*Latter study showed the presence of elevated aerosol during monsoon season and wet scavenging is clearly brought out.*

**Unfortunately, the instrumentation used for Figure 3 does not show useful data for the monsoon period, due to the monsoonal system and heavy cloud cover present over the study region. This caused noisy data that does not clearly show the aerosol present and its structure. Therefore, only the two days in the pre-monsoon has been included in our study. We have now included these two references though in the discussion section in order to highlight agreement between our aerosol structure measured in the pre-monsoon and monsoon seasons, as of Sinha et al (2013) and Ratnam et al (2018).**

*3. Elevated aerosol layers over Asian region are formed either due to convective or long-range transport. If it is through convective transport, chemical composition near surface should match with that observed aloft. In this study, surface concentrations of the chemical composition of the aerosol are missing. Information from either earlier published literature or their own surface measurements will be useful in interpreting the role of convective and long-range transport. I am wondering whether surface concentrations are measured with same set of instruments onboard aircraft before starting of the aircraft measurements each time?*

**There are no surface measurements with the same set of instruments before starting the aircraft measurements each time. Due to operational restrictions, we were unfortunately unable to measure any surface concentrations prior to each flight.**

*4. One of the conclusions that the dust and sulphate dominated aerosol layer aloft was removed upon monsoon arrival matches with the findings of Vernier et al. (2018) where they found Nitrate as dominant source in the UTLS region using zero pressure balloons from Hyderabad whereas sulphate near the surface. Some of these finds are useful in further interpreting the results.*

**Thank you for highlighting this useful reference (Vernier et al, 2018) to further interpret our results. Their findings show consistencies with our aerosol layer aloft being removed with monsoon arrival, and this reference has been added to the discussion section (page 13, line 19).**

*5. Are the profiles in figure 4 is the average of two flights in each season? If yes, it is better to show them separately to feel how different these profiles are within the same season (one can add as supplementary figure, if you feel that already figures are more).*

**The profiles in Figure 4 are averages of each season, for pre-monsoon this is 2 flights (B956 and B957), and the monsoon average from 8 flights (B968-B976). The flight profiles are separated later in Figure 10 so that the reader can get a sense of how different the profiles are both within both the same season, and between the two seasons.**

*6. Page 11, line 7: It is mentioned that elevated aerosol layer diminish as monsoon arrives leaving aerosol only within boundary layer. If monsoon washout (wet scavenging) is expected to remove the elevated aerosol layer, then throughout the profile including boundary layer aerosol should have also washed out? Why only elevated aerosol layer is diminished? Further there was no mention of rainout process which is also important during monsoon.*

**Noted – the text has been edited to clarify the various processes at hand that are influencing the aerosol loading as the monsoon progresses over northern India. It is beyond the scope of our paper to ascertain a relative contribution to the effect and influence of the various processes removing aerosol, but our paper highlights the complex nature of the monsoon system. As the monsoon arrives over northern India, Figure 2 shows a change in direction of air mass direction for the boundary layer and aloft. This change in air-mass direction coincides with a reduction in the total aerosol mass concentration throughout the profile, especially aloft where the aerosol we measured is almost all removed. This process is of course coincident with rainfall arrival. As suggested in our paper, the boundary layer aerosol removal may be less significant may be due to the strength of emissions across northern India, especially inside the IGP, effectively recharging the aerosols in that layer. If the aerosol is removed from the free troposphere there may be no replacement mechanism, whereas there is inside the BL, therefore providing a driver to maintain significant aerosol mass concentrations inside the BL.**

*7. There was no mention of true boundary layer altitude though many times it is used. 950 hPa do not represent throughout the Boundary layer though it is expected that within boundary layer all the spices are well mixed. In figure 3 description, boundary layer is considered as 2-4 km?*

**In our paper, the use of boundary layer height is to provide a general indication of the boundary layer height. The 950 hPa altitude was used as for most locations across northern India this was within the boundary layer, in order to provide an indication of the average air-flow direction in the pre-monsoon and monsoon seasons. We appreciate that the boundary layer varies between regions, but the use of 950 hPa should be safely within the boundary layer in all regions considered.**

*8. As also pointed out by other reviewer, use of CALIPSO profiles (aerosol, extinction and aerosol types) will be very useful for two reasons. One is to know how good satellite retrievals over this region and also whether aerosol types obtained from CALIPSO match with aircraft observations or not.*

**As explained following the comments from referee 1, this information has now been added to the paper. The CALIPSO satellite retrieval from the daytime pass on June 11$^{th}$ has been added and is Figure 4. The CALIPSO data shows useful comparison and agreement with the on-board lidar plot in Figure 3. This consistency between multiple data sources strengthens our findings.**

**MINOR COMMENTS**

*There is repeation in the abstract (lines 17 and 24) ...that total mass reduced to 50%*

**Complied with – we have removed the repetition from the text (line 24).**

*Page 8 line 6: How average total mass concentrations are obtained is not clear.*

**Noted – we have added in brackets that the average total mass concentrations are obtained by totalling the AMS aerosol species (Organics, Sulphate, Ammonium, Nitrate) and BC mass concentration. The "average" refers to the average across either inside the IGP or outside IGP region.**

*Page 13, line 16: Quoted reference over China is almost a decade back where level of pollution over India may also be less at that time. Re-wording of this statement is required.*

**This has been taken into account, so we have reinforced the statement with a more recent reference consistent with that of Ho et al (2006); Ni et al (2018) present similar ratios for China in summer seasons.**

*Additional References:*

*Sinha, P. R., Manchanda, R.K., Kaskaoutis, D.G., Kumar, Y.B., and Sreenivasan, S. (2013). Seasonal variation of surface and vertical proïﹾ‚Ale of aerosol properties over a tropical urban station Hyderabad, India. J. Geophys. Res. 118, doi:10.1029/2012JD018039.*

**Included now.**

*Vernier, J., et al. (2018). BATAL: The Balloon measurement campaigns of the Asian Tropopause Aerosol Layer. Bull. Amer. Meteor. Soc., https://doi.org/10.1175/BAMSD-17-0014.1.*

**Included now.**

*Venkat Ratnam, M., P. Prasad, M. Roja Raman, V. Ravikirana, S.V.B. Rao, B.V. Krishna Murthy and A. Jayaraman. (2018). Role of dynamics on the formation and maintenance of the elevated aerosol layer during monsoon season over south-east peninsular India. Atmospheric Environment. 188. 10.1016/j.atmosenv.2018.06.023.*

**Included now.**

**Vertical and horizontal distribution of sub-micron aerosol chemical composition and physical characteristics across Northern India during pre-monsoon and monsoon seasons**

James Brooks[1], James D. Allan[1,2], Paul I. Williams[1,2], Dantong Liu[3], Cathryn Fox[4], Jim Haywood[4,5], Justin M. Langridge[4], Ellie J. Highwood[6], Sobhan K. Kompalli[7], Debbie O'Sullivan[4], S. Suresh Babu[7], Sreedharan K. Satheesh[8], Andrew G. Turner[2,6], Hugh Coe[1].

[1] Centre for Atmospheric Science, School of Earth and Environmental Sciences, University of Manchester, Manchester, UK.

[2] National Centre for Atmospheric Science, UK.

[3] School of Earth Sciences, Zhejiang University, China.

[4] Observation Based Research, Met Office, Exeter, UK.

[5] College of Engineering, Mathematics & Physical Sciences, Exeter, UK.

[6] Department of Meteorology, University of Reading, UK.

[7] Space Physics Laboratory, Vikram Sarabhai Space Centre, India.

[8] Centre for Atmospheric & Oceanic Sciences, Indian Institute of Science, India.

*Correspondence to*: Prof. Hugh Coe (hugh.coe@manchester.ac.uk)

**Abstract.**

The vertical distribution in the physical and chemical properties of submicron aerosol has been characterised across northern India for the first time using airborne in-situ measurements. This study focusses primarily on the Indo-Gangetic Plain, a low-lying area in the north of India which commonly experiences high aerosol mass concentrations prior to the monsoon season. Data presented are from the UK Facility for Airborne Atmospheric Measurements BAe-146 research aircraft that performed flights in the region during the 2016 pre-monsoon (11th and 12th June) and monsoon (30th June to 11th July) seasons.

Inside the Indo-Gangetic Plain boundary layer, organic matter dominated the submicron aerosol mass (43%) followed by sulphate (29%), ammonium (14%), nitrate (7%) and black carbon (7%). However, outside the Indo-Gangetic Plain, sulphate was the dominant species contributing 44% to the total submicron aerosol mass in the boundary layer, followed by organic matter (30%), ammonium (14%), nitrate (6%) and black carbon (6%). Chlorine mass concentrations were negligible throughout the campaign. Black carbon mass concentrations were higher inside the Indo-Gangetic Plain (2 µg/m$^3$) compared to outside (1 µg/m$^3$). Nitrate appeared to be controlled by thermodynamic processes, with increased mass concentration in conditions of lower temperature and higher relative humidity. Increased mass and number concentrations were observed inside the Indo-Gangetic Plain and the aerosol was more absorbing in this region, whereas outside the Indo-Gangetic Plain the aerosol was larger in size and more scattering in nature, suggesting greater dust presence especially in northwest India. The aerosol composition remained largely similar as the monsoon season progressed, but the total aerosol mass concentrations decreased by ~50% as the rainfall arrived; the pre-monsoon average total mass concentration was 30 µg/m$^3$ compared to a monsoon average total mass concentration of 10-20 µg/m$^3$. However, this mass concentration decrease was less noteworthy (~20-30%) over the Indo-Gangetic Plain, likely due to the strength of emission sources in this region. Decreases occurred in coarse mode aerosol, with the fine mode fraction increasing with monsoon arrival. In the aerosol vertical profile, inside the Indo-Gangetic Plain during the pre-monsoon, organic aerosol and absorbing aerosol species dominated in the lower atmosphere (<1.5 km) with sulphate, dust and other scattering aerosol species enhanced in an elevated aerosol layer above 1.5 km with maximum aerosol height ~6 km. The elevated concentration of dust at altitudes >1.5 km is a clear indication of dust transport from the Great Indian Desert, also called the Thar Desert, in north-western India. 
[revised manuscript text omitted]

15 measurements show peak concentration between 1.5 to 2 km for June 11th flight B956, whereas for B957 on June 12th shows centroid of the aerosol layer at 3km with the presence of aerosols with reduced extinction from 6 km to near-surface. The thermodynamic profiles in Figure 3 for the various locations show differences between the boundary layer and aloft across northern India; heights that coincide with elevated aerosol have consistent high RH values across locations, with large variations in RH in the boundary layer. It is clear from the lidar data that there is distinct structure in aerosol vertical profiles.
20 The previous features are also complemented in Figure 4, which is a CALIPSO aerosol backscatter profile from a daytime overpass from 11th June (B956). The aerosol structure is comparable between both lidar plots, with the CALIPSO successfully capturing the maximum aerosol height during the pre-monsoon in northern India (~6-7 km) along with the values shown representing dust aerosol dominance, consistent with the literature for the pre-monsoon season. The CALIPSO profile does not, however, successfully highlight the distinct structure in the aerosol chemical and physical properties.

[revised manuscript text omitted]